# DynamX Bioadaptor as an Emerging and Promising Innovation in Interventional Cardiology

**DOI:** 10.3390/life15101549

**Published:** 2025-10-02

**Authors:** Julia Soczyńska, Kamila Butyńska, Mateusz Dudek, Wiktor Gawełczyk, Sławomir Woźniak, Piotr Gajewski

**Affiliations:** 1Student Scientific Group of Heart Diseases, Wroclaw Medical University, 50-556 Wrocław, Poland; kamila.butynska@student.umw.edu.pl (K.B.); mateusz.dudek@student.umw.edu.pl (M.D.); wiktor.gawelczyk@student.umw.edu.pl (W.G.); 2Division of Anatomy, Department of Human Morphology and Embryology, Wroclaw Medical University, 50-556 Wroclaw, Poland; slawomir.wozniak@umw.edu.pl; 3Institute of Heart Diseases, Wroclaw Medical University, 50-556 Wrocław, Poland; p.gajewski@umw.edu.pl

**Keywords:** DynamX Bioadaptor, coronary artery disease, interventional cardiology, stent, clinical outcomes

## Abstract

Coronary artery disease (CAD) remains a major cause of mortality worldwide. Among the standard therapeutic approaches are percutaneous coronary interventions (PCI) employing stents. The main limitation of the procedure lies in the permanent stiffening of the vessel wall. The DynamX Bioadaptor, representing a new generation of vascular stents, combines the advantages of standard implants with a unique mechanism—“uncaging.” Its helical structure, linked by a biodegradable material, enables the restoration of the vessel’s natural functions. This breakthrough concept in interventional cardiology holds the potential to establish a new standard of care for patients suffering from CAD. In this work, we aim to synthesize the available evidence concerning the characteristics of the DynamX Bioadaptor and its impact on vascular physiology. We provide a comprehensive review and evaluation of current clinical reports on its use, analyzing the available literature in comparison with other stent technologies. Recognizing that the DynamX Bioadaptor is a relatively recent innovation, we also seek to identify existing gaps in the literature and propose future directions for research to fully assess its long-term clinical potential.

## 1. Introduction

CAD remains a significant challenge in contemporary medicine. Evidence indicates that it ranks high among the leading causes of death worldwide, accounting for nearly half of all cardiovascular disease-related fatalities [1,2]. It is important to emphasize that CAD may be associated with a range of complications. Reported complications include transient arrhythmias or arrhythmias requiring immediate medical intervention, mitral valve insufficiency, and structural damages such as rupture of the papillary muscle [3]. Attention should also be drawn to a significant complication, namely heart failure [4]. In clinical practice, early diagnosis and the planning of a therapeutic strategy are crucial to prevent deterioration of the patient’s condition [5]. Statistics show that mortality due to this condition is increasing most rapidly among older individuals, and demographic aging is expected to contribute to a higher prevalence of CAD in the future [6,7]. The pathophysiology is primarily based on the obstruction or impairment of blood flow in the coronary vessels, mainly due to the accumulation of lipid material in the intimal layer, accompanied by inflammation, particularly in chronic settings. The resulting structure, known as an atherosclerotic plaque, may be prone to erosion or rupture, leading to significant clinical consequences [8,9]. Particular attention should be paid to this disease in the context of the COVID-19 pandemic, which has been associated with an increased incidence of acute coronary syndromes, such as ST-segment elevation myocardial infarction. Although COVID-19 is not the primary focus of this study, it is worth noting its association with cardiac patients, as this may have implications for planning revascularization strategies. Consequently, the importance of early diagnosis and effective, rapid treatment of CAD is heightened, including in patients affected by SARS-CoV-2 [10,11]. The significance of CAD and the necessity for timely and effective management is further supported by the 2024 update of the European Society of Cardiology guidelines [12]. Treatment, alongside supportive diet and rehabilitation, relies on pharmacological therapy and interventional methods [13,14]. Regarding interventional therapy, its history dates back to the 1960s with the introduction of coronary artery bypass grafting. By 1978, advances enabled PCI [15]. Today, approximately 90% of PCI involve the use of stents—small tubular scaffolds classified according to material, expansion method, and structural design. The main categories include: (1) BMS, introduced in the 1980s; (2) DES, implemented at the beginning of the 21st century; and (3) BVS [16,17]. A more detailed discussion of these devices is provided in a dedicated section. Stents are now regarded as life-saving innovations and are rightly considered among the most significant medical achievements of the century. However, despite their functionality, they are not free from complications, including thrombosis and restenosis. The latter, associated with stent collapse and altered target vessel diameter, also depends on clinical presentation and scaffold type. These challenges prompted the development of bioresorbable stents, designed to reduce such complications [18,19]. Similar concerns arose regarding structural issues during the resorption process [17]. Given certain limitations, such as suboptimal vascular remodeling, the concept of a device with a mechanism functioning transiently, similar to a DES, but integrated with features allowing long-term adaptive vessel responses, became particularly relevant. In recent years, much attention has focused on the DynamX Bioadaptor™ (Elixir Medical Corporation, Milpitas, CA, USA), which combines these characteristics and appears to have potential for favorable vascular effects [20].

This review aims to systematize the available evidence regarding the characteristics of the DynamX Bioadaptor and its impact on the natural functions of blood vessels in the context of contemporary interventional cardiology. We highlight its unique features compared with other stent technologies, provide a comprehensive review of the available clinical evidence, and assess the advantages and limitations reported in clinical studies. Recognizing that the Bioadaptor represents a relatively new technology, we also seek to identify directions for future research.

## 2. Classification of Stents Based on Material

Stents, also referred to as tubular implants, serve to restore patency in narrowed arteries or other conduits by providing mechanical support and preventing re-occlusion. Based on their mechanism of action, stents are classified as either balloon-expandable or self-expanding. The first generation of stents was predominantly manufactured from metals and is therefore referred to BMS [21]. An appropriate metallic stent should exhibit the following characteristics: corrosion resistance, low thrombogenicity, high flexibility, sufficient radiopacity for precise fluoroscopic guidance, and adequate radial strength. These features are essential for the proper functioning of such stents. Biocompatibility is also a critical parameter, as it reduces the risk of neointimal hyperplasia (NIH) and thrombosis during long-term follow-up [22]. Initially, BMS were made from stainless steel; however, metal alloys such as cobalt-chromium or platinum-chromium began to replace it due to their superior strength [16,22]. Alternative materials include alloys of tantalum, platinum, and niobium [23].

Another distinct category of stents is DES, which consist of a metallic scaffold, a polymeric coating (if present), and an antiproliferative drug. Similarly to BMS, the scaffold of DES is typically constructed from stainless steel or metal alloys [24]. The most recent scaffolds are composed of cobalt-chromium and platinum-chromium alloys, both of which exhibit higher tensile strength than stainless steel [24,25]. The polymeric coating on the stent surface contributes to drug-release modulation, prevents premature washout of the drug, and enhances biocompatibility [26]. Synthetic polymers such as poly(n-butyl methacrylate) (PBMA) and poly(ethylene-co-vinyl acetate) (PEVA) are known to have been employed by the first generation of DES. Due to the potential inflammatory response triggered by synthetic polymers, second-generation DES were developed with more biocompatible polymers, such as poly(vinylidene fluoride-co-hexafluoropropylene) (PVDF-HFP) and phosphorylcholine (PC), a component of the phospholipid bilayer of cell membranes. Antiproliferative drugs used in first-generation DES included sirolimus and paclitaxel, whereas later generations employ sirolimus derivatives such as everolimus (with anti-atherosclerotic properties), novolimus, and zotarolimus [25].

Drug-coated balloons (DCBs) have emerged as an alternative to DES, typically utilizing drugs such as sirolimus and paclitaxel. The purpose of DCB is to deliver the drug to the arterial wall endothelium. In addition to the active drug, DCB includes a hydrophilic, inert excipient that facilitates adherence and dissolution of the drug within the arterial wall [27]. Unlike DES, DCB enables uniform drug application to the vessel wall without the need for implantation of a metallic stent [28].

Biodegradable stents (BRSs) represent another class, which, unlike earlier-generation stents, undergo biodegradation and resorption. This allows arterial to maintain early patency and physiological contractility while reducing the risk of late complications [29,30]. BRS are constructed from absorbable materials that provide high flexibility and biocompatibility, minimizing the risk of scaffold damage during implantation and allowing limited elastic recoil during subsequent phases [31]. Ideal BRS should exert sufficient radial force on the vessel to prevent restenosis for approximately six months [32]. Most BRS are based on polylactic acid polymers, iron alloys, tyrosine copolymers, and magnesium alloys; however, the latter exhibit high corrosion rates, leading to complete degradation within three months post-implantation [32,33].

## 3. General Characteristics of the DynamX Bioadaptor Device

The DynamX Bioadaptor is an advanced vascular implant eluting sirolimus, used in the treatment of ischemic heart disease. It was designed to overcome the limitations of conventional DES, which permanently rigidify the vessel after implantation, restricting its mobility and physiological function. Its uniqueness compared to other types of stents lies in its innovative construction [34,35]. The DynamX Bioadaptor consists of a thin (71 μm) cobalt-chromium scaffold and sinusoidal rings equipped with drug-eluting elements [36]. Approximately six months after implantation, polymer degradation occurs, leading to the restoration of the vessel’s physiological functions, such as adaptive remodeling, cyclic pulsation, vascular motion, and functional accommodation to the vascular structure [35]. By allowing the artery to be freed from the rigid stent cage and restoring its natural mobility and function, this device may potentially contribute to a reduction in late clinical complications [35,37].

## 4. Differences Compared to Current Technologies

Currently, the implantation of DES is considered the gold standard in PCI. Over the years, these stents have undergone significant improvements, such as the reduction in the thickness of the wires constituting the stent. The result of these modifications—including the reduction in periprocedural complications, stent thrombosis, and restenosis—has contributed to the enhancement of both the efficacy and safety of PCI [38]. Unfortunately, despite these advances, DES still create a rigid scaffold within the coronary artery into which they are implanted [39]. Arterial scaffolding can result in stretching of the vessel, torsion, and inhibition of adaptive remodeling [40]. Furthermore, leaving a metallic scaffold in the vessel may initiate a chronic inflammatory state, leading to NIH. Stent implantation may cause migration of myofibroblasts and smooth muscle cells, resulting in the formation of neointima and, consequently, NIH. It is assumed that this mechanism may occur in both DES and BRS. In the case of BRS, “dismantling” may occur, which can increase inflammation and the risk of late scaffold thrombosis. Factors such as advanced age, diabetes, reduced ejection fraction, discontinuation of dual antiplatelet therapy, acute coronary syndrome, chronic kidney disease, and smoking may be associated with BRS failure, which in turn may lead to stent restenosis and stent thrombosis [41]. What distinguishes the DynamX bioadaptor from earlier-generation stents is the use of the “uncaging” mechanism, which, after resorption of scaffold elements, partially restores vessel elasticity and improves its pulsatility. This process may lead to a reduction in proliferative and inflammatory responses [42]. These factors can impair hemodynamics, which may result in complications such as stent damage, stent thrombosis, myocardial infarction, and restenosis, often necessitating repeat revascularization [43,44]. Due to the numerous limitations associated with the permanent structure of DES, bioresorbable scaffolds were developed, as described previously. Their aim was to combine the short-term benefits of permanent stent implantation with the possibility of gradual scaffold resorption, thereby enabling the restoration of physiological vessel function and mobility [33]. According to Forrestal et al., bioresorbable scaffolds were intended to “leave nothing behind” protecting against chronic inflammation, permitting uninterrupted future vascular imaging, and preserving distal bypass graft sites. However, following the publication of studies comparing bioresorbable scaffolds with DES, accumulated data revealed significant limitations and disadvantages associated with scaffold implantation. Pooled analyses of the ABSORB trials with three-year follow-up demonstrated that, compared with DES, bioresorbable scaffolds were associated with increased rates of target lesion revascularization, both early and late scaffold thrombosis, and a higher risk of target-vessel myocardial infarction [33,45,46]. The ABSORB III trial demonstrated the following outcomes at five-year follow-up: target lesion failure (TLF) was observed in 17.5% of patients treated with bioresorbable scaffolds compared with 15.2% in the DES group (*p* = 0.15), along with higher rates of scaffold thrombosis (2.5% vs. 1.1%; *p* = 0.03) and target-vessel myocardial infarction (10.4% vs. 7.5%; *p* = 0.04) [46]. Studies also observed scaffold dxuption, uncontrolled scaffold degradation [47], and significantly greater late lumen loss (LLL) for bioresorbable scaffolds compared with BMS [48]. In patients with in-stent restenosis, the European Society of Cardiology recommends as first-line therapy either the implantation of a new-generation DES (class I, level A) or a DCB. Meta-analyses, however, indicate some discrepancies between these strategies. In the study by Cai et al., it was found that, regardless of the type of in-stent restenosis, new-generation DES may lead to better short-term clinical and angiographic outcomes compared with DCB [49]. More recent studies, however, have demonstrated that DCB are associated with higher rates of restenosis compared with DES and should be limited to carefully selected, uncomplicated lesions [34]. Another limitation of DCB is the short duration of drug delivery to the vessel, which is strictly determined by the balloon inflation time, resulting in less than 10% of the drug reaching the arterial wall [50]. Nevertheless, they can be employed in combination with DES, reducing the total stent length used while simultaneously lowering the risk of restenosis, particularly in more complex cases [51,52]. Figure 1 and Figure 2 illustrate the structural differences between conventional stents and the DynamX Bioadaptor.

## 5. Restoration of Physiological Vascular Functions and Mechanism of Action

### 5.1. Restoration of Vascular Pulsatility

The DynamX Bioadaptor enables the restoration of coronary artery pulsatility during myocardial contraction in the treated segment [53]. In the BIOADAPTOR randomized controlled trial, the lumen area of the treated segment increased by 7.5% between systole and diastole, yielding results comparable to untreated vessel segments, whereas treated with DES remained rigid. This translates into a reduction in the functional workload difference between treated and untreated segments, an effect unattainable with conventional stents. Consequently, this reduces the risk of flow disturbances within the artery and edge restenosis. In an intravascular ultrasound (IVUS) study, the maximum lumen area increased by 46% at 9 to 12 months post-procedure. Scaffold ring removal also enhanced the vessel’s responsiveness to nitroglycerin, which increased from 0.03 mm^2^ immediately after implantation to 0.17 mm^2^ at follow-up [42].

### 5.2. Positive Remodeling and Lumen Expansion

The described Bioadaptor exerts a beneficial effect on coronary artery remodeling. In contrast to DESs, which cause a progressive reduction in the lumen diameter achieved during the procedure, the DynamX Bioadaptor allows for its preservation [35]. This occurs despite complete endothelial coverage of the scaffold, as demonstrated using Optical Coherence Tomography (OCT). The endothelial layer covering the Bioadaptor six months post-implantation is comparable to that typically covering DES (100 ± 60 μm vs. 101.7 ± 65.4 μm) and covers 96% of the struts without causing scaffold displacement [35,54]. Moreover, after Bioadaptor implantation, a 3% increase in the mean vessel lumen area and a 5% increase in the mean device area were observed, enabling the maintenance of a stable diameter and the artery’s adaptation to neointimal growth [36].

In preclinical animal studies using OCT, it was observed that three months post-procedure, the lumen diameter decreased slightly due to local endothelial proliferation; however, once scaffold delamination occurred, the diameter increased at 12 and 24 months, indicating favorable remodeling. This finding was confirmed in a clinical study involving 18 patients, where the mean lumen cross-sectional area measured by IVUS increased from 7.22 mm^2^ immediately post-implantation to 7.32 mm^2^ at 9–12 months in segments treated with the Bioadaptor. The study further reported that the DynamX Bioadaptor restores the vessel’s physiological axis: after scaffold delamination, the axis changed from 137.6 ± 16.2° immediately post-procedure to 157.5 ± 14.5° post-implantation and then stabilized at 149.7 ± 16.1° at the 9–12-month follow-up, as assessed by coronary angiography [42].

### 5.3. Impact on Vascular Wall Biology

Smooth muscle cell proliferation in vessels following injury is regulated by specific genes and the control of their transcription. Genes such as ACTA2, MYH11, SM22α, and SM-MHC are responsible for the process of vascular regeneration. In samples obtained from coronary arteries at three and six months post-procedure, no significant differences were observed. However, at nine months, vessels implanted with the DynamX Bioadaptor exhibited increased expression of these genes compared with samples from these treated with Xience DES, indicating accelerated healing and potentially contributing to a lower incidence of complications. Gene expression in the case of DES implantation reached comparable levels to DynamX only three months later, twelve months post-procedure [42,55]. Furthermore, after scaffold delamination, peak stresses within the Bioadaptor were reduced by 70%, lowering the risk of vascular inflammation, stent damage, and associated late post-procedural complications. Bioadaptors simultaneously preserve arterial mobility, reducing mechanical stress imposed by the stent and mitigating adverse outcomes such as stent fracture [56]. A 36-month analysis of the BIOADAPTOR randomized controlled trial demonstrated that no thrombotic events, including stent thrombosis or target-vessel myocardial infarction, occurred in patients with the Bioadaptor. This rate is more favorable compared with contemporary DES, where the stent thrombosis rate ranged between 0.5% and 1.4% in the TALENT, BIONYX, and BIO-RESORT trials [42].

### 5.4. Impact on Atherosclerotic Plaque Volume

In arteries implanted with the Bioadaptor, segments containing the device demonstrated stabilization of atherosclerotic plaque volume at 12-month follow-up, whereas plaques in these treated with DES exhibited an increase in volume (3% increase with DynamX vs. 12% increase with DES). When plaques containing calcium were excluded from the analysis, the Bioadaptor was associated with regression of atherosclerotic plaque volume, particularly in lipid-rich lesions (−9% vs. 10% with DES) and non-calcified lesions (−4% vs. 9% with DES), whereas plaques in DES-treated vessels continued to enlarge. This effect is thought to result from the synergistic restoration of physiological vessel function, pulsatility, and mobility, combined with lipid-lowering therapies. No differences were observed between patient groups in plaque volume changes within vascular segments proximal or distal to the implant [40,54].

### 5.5. Significance for High-Risk Patients

Long lesions in vascular segments require the implantation of extended stents. In the case of DES, this further restricts vessel mobility and increases the risk of mechanical injury, effects that can be mitigated by the use of the DynamX Bioadaptor, which provides support in this context. The Bioadaptor may also be more effective in restoring normal physiological function in the left anterior descending artery (LAD) post-treatment, as this artery is particularly prone to atherosclerotic changes and is subject to pulsatile motion during cardiac contraction. Observations indicating lower LLL with DynamX compared to DES further suggest that its use may be especially advantageous for patients at elevated risk of restenosis, including those with diabetes [34,57].

A summary of the above is presented in Figure 3.

## 6. Clinical Trial Results of the DynamX Bioadaptor

A 12-month analysis of the BIOADAPTOR randomized controlled trial demonstrated that DynamX was non-inferior to Resolute Onyx in terms of the incidence of TLF (1.8% [n = 4; 95% CI: 0.5–4.6] vs. 2.8% [n = 6; 95% CI: 1.0–6.0]; *p*_non < 0.001 for non-inferiority). The same study, comparing the efficacy of DynamX with DES, also revealed advantages of DynamX in specific patient subgroups, particularly those with lesions in the LAD and long lesions (≥23 mm). Compared with Resolute Onyx DES, DynamX demonstrated superior LLL and a lower percentage of luminal narrowing. In small vessels (≤2.75 mm), patients treated with DynamX similarly exhibited a reduced rate of diameter stenosis. Moreover, IVUS analyses showed that patients treated with DynamX had a smaller NIH volume compared with those receiving DES. At 12-month follow-up, vessel stenosis in the DynamX group was 12.7% versus 17.3% for DES (*p* = 0.05), and LLL was 0.09 mm for DynamX compared with 0.25 mm for DES (*p* = 0.038), demonstrating a significant advantage of DynamX in this regard. Procedural success and the acute gain in lumen diameter after DynamX implantation were comparable to those achieved with DES [54]. Two-year follow-up confirmed significant clinical benefits for patients, suggesting greater efficacy of the Bioadaptor in treating ischemic heart disease. The TLF rate was lower with DynamX than with DES (1.8% vs. 5.5%; log-rank *p* = 0.044), which may be attributed to arterial support provided by the Bioadaptor after scaffold delamination [40]. Similarly, the six-month INFINITY-SWEDEHEART study demonstrated clinically meaningful benefits in the Bioadaptor group compared with DES, including lower rates of target vessel failure, TLF, ischemia-driven target lesion revascularization, and myocardial infarction [58]. The single-center study described by Wong et al. allowed for conclusions regarding the positive impact of the DynamX Bioadaptor in STEMI patients undergoing primary percutaneous coronary interventions. The procedure was deemed safe, and favorable outcomes at one-year follow-up were confirmed [59].

In summary, the Bioadaptor represents a solution that combines the procedural efficacy of DES with a modern scaffold-delamination mechanism, enabling positive vessel remodeling and restoration of physiological coronary artery function [60].

A brief comparison of DES and the DynamX Bioadaptor is presented in Table 1.

## 7. Discussion

Due to its unique properties, the DynamX Bioadaptor is undoubtedly garnering increasing interest within the medical community regarding interventional cardiology procedures. Despite its key advantages, according to the most recent guidelines, the Bioadaptor is currently not indicated as a preferred tool within standard treatment protocols [12]. Its expanding potential, however, merits attention. It is plausible that ongoing and future clinical studies may contribute to the redefinition of core therapeutic strategies and lead to the establishment of new treatment paradigms, analogous to the historical transition from BMS to DES due to superior outcomes [61] and the subsequent recommendation of DES as standard therapy for patients with CAD [62]. A study conducted by Saito et al. indicated that the DynamX Bioadaptor appears to be as safe as, or potentially safer than, conventional DES. These conclusions were based on clinical event analysis in a cohort of 440 patients, revealing a single repeat revascularization in the DynamX group versus four in the DES group, alongside two cardiovascular deaths [40]. It is reasonable to hypothesize that the device may particularly benefit younger patients, for whom the long-term risk of restenosis and re-intervention is clinically significant, and the DynamX Bioadaptor could also potentially play a role in the treatment of de novo lesions, where a favorable adaptive response may be achievable, although those hypotheses require further validation. Potential limitations are associated with technical challenges. It is crucial to determine whether, in the long-term perspective, there are any adverse effects related to the interaction mechanism of the metallic segments and whether the observed effects are sustained over time. Considering factors contributing to stent thrombosis—such as acute events, suboptimal adherence to medical recommendations, and comorbidities—clarifying preventative strategies in younger populations is essential [63]. Overall, the device appears promising for maintaining long-term vessel patency and may represent an attractive option for younger patients with longer expected life spans, though this must be confirmed through clinical trials. Currently, there are no reports in the literature regarding the clinical outcomes of the DynamX Bioadaptor specifically in younger patient populations; existing studies focus predominantly on populations with a mean age of approximately 60 years [35,64,65]. Another gap in the literature concerns long-term outcomes: current studies have observation periods of up to five years, such as the single-blind, randomized INFINITY-SWEDEHEART trial with 2400 participants [34], or the BIOADAPTOR randomized controlled trial [54]. Open questions remain regarding the structural durability of the device and potential alterations in its architecture due to segmental micromovements, as well as endothelial responses over the long term. The DynamX Mechanistic Study, a prospective, multicenter, non-randomized study, provides follow-up data for a maximum of 36 months [42]. The available evidence has limitations. For instance, as Capodanno noted regarding the INFINITY-SWEDEHEART randomized controlled trial, the study population had a limited burden of comorbidities and a simpler CAD profile, potentially influencing observed outcomes [66]. Attention must also be given to the relatively small number of studies, the limited reporting of case reports, and the modest size of study populations. New studies are planned to address these limitations. An example is the ongoing international BIO-RESTORE registry, initiated in 2024, aiming to enroll 5000 participants to evaluate safety and efficacy profiles in patients with CAD; results have not yet been published. Differences in study design and patient populations should be noted. The global registry includes a broad cohort of patients undergoing percutaneous coronary intervention with the DynamX Bioadaptor, with minimal exclusion criteria, such as target lesion location in the left main coronary artery, prior grafting, or restenosis [67]. INFINITY-SWEDEHEART recruits patients with ischemic heart disease and de novo lesions in previously unaffected coronary arteries to compare the safety profile of the DynamX Bioadaptor with Resolute Onyx [34], whereas the BIOADAPTOR randomized controlled trial evaluates clinical outcomes in patients with de novo lesions treated with the DynamX Bioadaptor versus DES [54]. Despite limitations such as selective clinical populations in the BIOADAPTOR RCT and geographically restricted recruitment in INFINITY-SWEDEHEART RCT, the studies complement one another, collectively contributing to a more comprehensive understanding of the DynamX Bioadaptor’s safety and efficacy relative to other stents. Ongoing research will be crucial to fill gaps in the literature, particularly regarding late stent thrombosis and post-procedural hemodynamic monitoring. Beyond clinical outcomes, economic considerations merit attention. To our knowledge, no direct cost analyses comparing the DynamX Bioadaptor to other stents have been published. Given the limited number of adverse events [40], the device may offer cost advantages in specific clinical contexts despite potentially higher device costs; however, dedicated economic studies are required to substantiate this theoretical benefit. Clinical adoption of the DynamX Bioadaptor remains geographically limited. The device has received CE marking but is not commercially available in the United States, although it has obtained Breakthrough Device designation from the U.S. Food and Drug Administration, expediting evaluation [68]. This innovative technology may pose economic challenges, particularly in regions with limited financial resources and underdeveloped technical infrastructure. Literature emphasizes that diagnostic and device innovations are often concentrated in high-income countries, yet quality deficits in medical services affect both lower-income countries and specific regions of developed countries [69]. According to World Health Organization assessments, up to 70% of innovations designed for high-income countries do not function as intended in low- and middle-income countries [70]. The unique features of the DynamX Bioadaptor could be particularly relevant in less developed healthcare systems, although implementation may be hindered by costs and the absence of standardized protocols. Operator training is another critical factor for successful device deployment. Current literature does not provide direct evidence on this aspect. Implantation of the DynamX Bioadaptor requires adherence to specific procedural guidelines. In the BIOADAPTOR-RCT, DES implantation protocols were followed [54], reducing the need for extensive retraining. According to Lin et al., who report a rare case of implant dislodgement, the unique design of the DynamX Bioadaptor should be carefully considered [71]. Nonetheless, a deeper investigation into the learning curve for this device is warranted.

In summary, the analysis suggests that the DynamX Bioadaptor has potential for broad clinical application. The technology’s innovativeness distinguishes the implant from other solutions. Given its novel mechanism, consideration may be given to using the DynamX Bioadaptor in patients for whom conventional DES are less effective, although further clinical studies are necessary. They may ultimately prove to be more advantageous than fully bioresorbable designs, while simultaneously addressing a gap where conventional DES have well-recognized limitations. It is essential to consider factors such as patient age, disease progression, and lesion location. Current limiting factors include observation duration and restricted device availability. Potential logistical challenges and the need to define precise clinical indications and patient subgroups—considering age, vascular anatomy, and overall health—are highlighted. Case reports detailing the device’s use in rare clinical scenarios are lacking. Future studies will be instrumental in clarifying the role of the DynamX Bioadaptor in interventional cardiology and determining whether it should be considered an alternative option or a dominant technology. In Table 2, we can see a summary of the findings from clinical trials.

## 8. Conclusions

The DynamX Bioadaptor represents a promising technology in interventional cardiology, particularly significant due to its potential for long-term vascular protection. The literature indicates efficacy comparable to that of conventional DES; however, studies in carefully selected patient populations and reports of individual cases remain limited. Future studies with longer follow-up and registry data will be essential to establish the device’s role in clinical practice and to fully assess its potential.

## Figures and Tables

**Figure 1 life-15-01549-f001:**
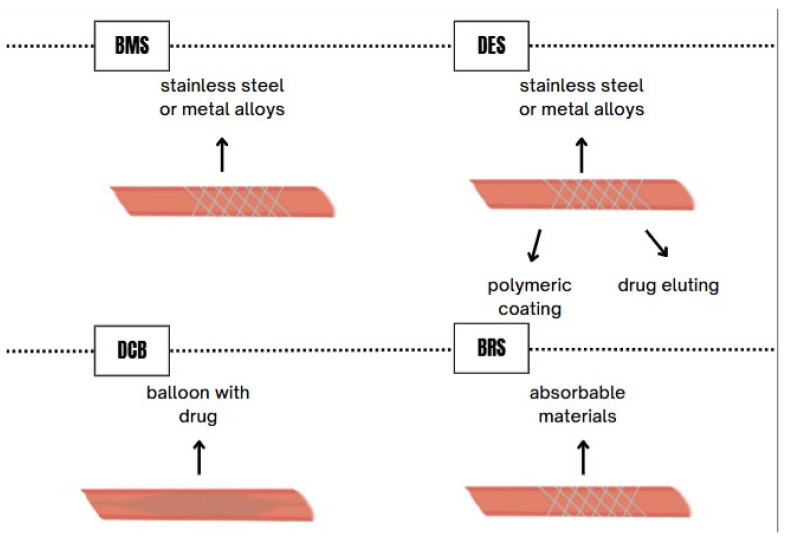
Structure of stents.

**Figure 2 life-15-01549-f002:**
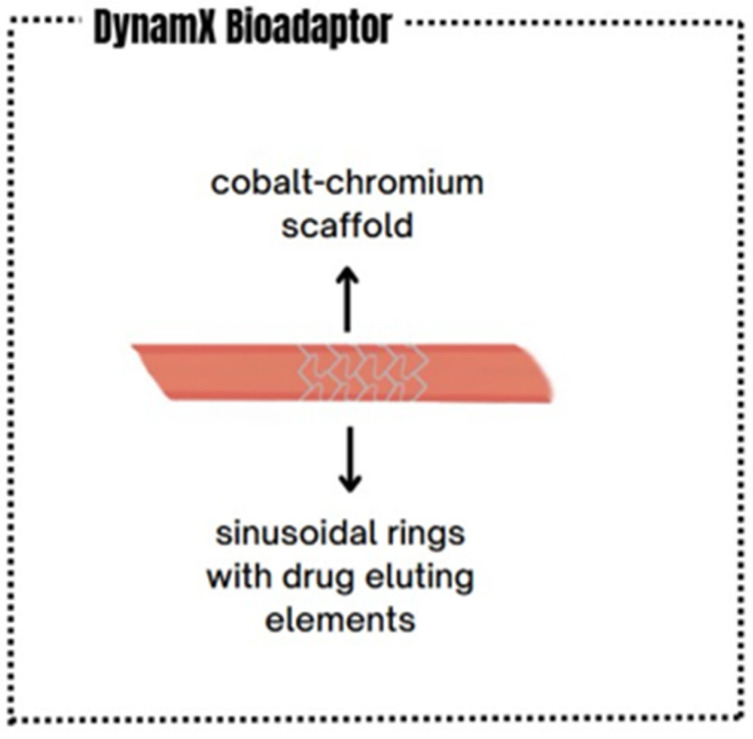
Structure of DynamX Bioadaptor.

**Figure 3 life-15-01549-f003:**
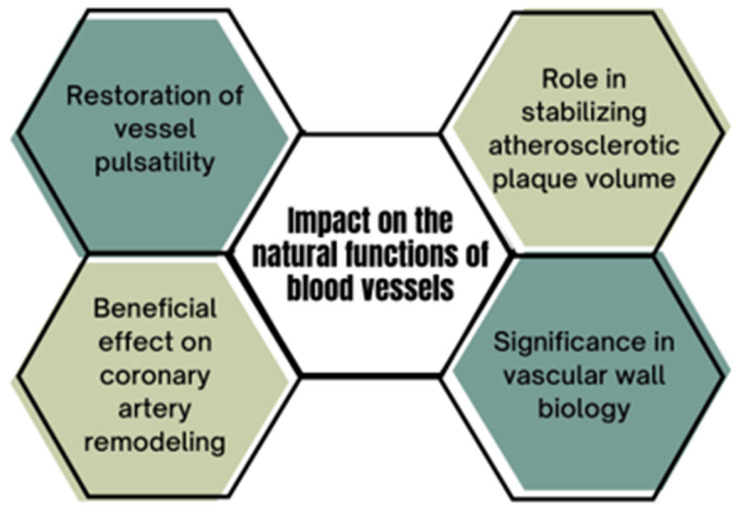
Impact of the DynamX Bioadaptor on the vascular system.

**Table 1 life-15-01549-t001:** A general comparison of DES and the DynamX Bioadaptor.

	DES	DynamX Bioadaptor
Construction	Permanent metal scaffold [24]	Stent with a de-caging mechanism [37]
Remodeling	Vessel caging, which may limit adaptive remodeling [40]	Allows for favorable remodeling [42]
Lumen Diameter	Reduction in the vessel lumen diameter over time [35]	Preservation of vessel lumen diameter even after endothelial coverage [35,54]

**Table 2 life-15-01549-t002:** Summary of the findings in clinical trials.

	Study	Year	n	Methods	Aim	Outcomes
1.	Saito et al. [54]	2023	445	Patients with coronary artery lesions, and without myocardial infarction in 34 hospitals were enrolled.	Clinical comparison between DES and Bioadaptor after 1 year.	DynamX presented similar effectiveness in terms of TLF, but also showed improved efficacy in parameters such as LLL, vesel function, and reduced risk of adverse events.
2.	Saito et al. [40] (follow-up)	2025	440	Patients with coronary artery lesions, and without myocardial infarction in 34 hospitals were enrolled.	Clinical comparison between DES and Bioadaptor after 2 years.	This follow-up confirmed benefits from using DynamX. Bioadaptor also showed lower TLF rate than DES.
3.	Erlinge et al. [34]	2024	2400	Patients with ischemic heart disease with coronary artery lesions from Sweden were enrolled.	Evaluation of safety and effectiveness of DynamX compared to Resolute Onyx stent.	DynamX demonstrated outstanding performance compared to DES, with lower rates of target vessel failure, target lesion failure, ischemia-driven target lesion revascularization, and myocardial infarction.
4.	Verheye et al. [42]	2023	50	Patients with coronary artery lesions from Belgium were enrolled.	Evaluation of safety and effectiveness of DynamX.	DynamX presented great clinical outcomes, such as absent of target-vessel myocardial infarction, and only one target lesion revascularization up to 36 months.

## Data Availability

The original contributions presented in this study are included in the article. Further inquiries can be directed to the corresponding authors.

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
