# Peer review of "DynamX Bioadaptor as an Emerging and Promising Innovation in Interventional Cardiology"

_life, 2025, doi:10.3390/life15101549_

Round 1
Reviewer 1 Report
Comments and Suggestions for Authors
-The link with the COVID-19 pandemic feels somewhat forced and could be reduced or better contextualized.
-In Page 8, in Table 1: “Dynam X Bioadaptor” there is incorrect spacing; should be “DynamX Bioadaptor.”
-The conclusions are appropriate but would benefit from a more balanced tone: instead of stating that DynamX represents a “significant innovative solution,” I suggest saying that it “represents a promising technology, with potential to be confirmed in larger studies with longer follow-up.”
Author Response
Comment 1: The link with the COVID-19 pandemic feels somewhat forced and could be reduced or better contextualized.
Response 1: Thank you for this valuable feedback. The section referring to the COVID-19 pandemic has been shortened and better contextualized. It now serves only as a brief mention, emphasizing the importance of early diagnosis and treatment of coronary artery disease in special clinical circumstances, without diverting attention from the main focus of the manuscript.
Comment 2: In Page 8, in Table 1: 'Dynam X Bioadaptor' there is incorrect spacing; should be 'DynamX Bioadaptor.
Response 2: Thank you for pointing this out. The typographical error has been corrected, and the proper name "DynamX Bioadaptor" is now consistently used throughout the manuscript, including Table 1.
Comment 3: The conclusions are appropriate but would benefit from a more balanced tone: instead of stating that DynamX represents a 'significant innovative solution,' I suggest saying that it 'represents a promising technology, with potential to be confirmed in larger studies with longer follow-up.
Response 3: We appreciate this suggestion. The conclusion section has been revised to reflect a more balanced tone."
Reviewer 2 Report
Comments and Suggestions for Authors
Dear Authors.
I congratulate to your work: it is concise and of great value and qulity.
Only 1 suggestion: pls add a figure on the device showing its structure possibly with the uncaging mechanism.
Sincerely
Author Response
Comment 1: Dear Authors. I congratulate you on your work: it is concise and of great value and quality.
Only 1 suggestion: please add a figure on the device showing its structure, possibly with the uncaging mechanism.
Response 1: Thank you very much for your kind words and valuable suggestion. In response, we have added new figures illustrating the stents and the DynamX Bioadaptor to improve clarity and visual understanding of the device’s structure.
Reviewer 3 Report
Comments and Suggestions for Authors
The topic of the review is highly relevant and relates to the new generation of intracoronary stents, which are designed to improve revascularization outcomes. There is insufficient information on this topic in the literature. The abstract fully reflects the content of this review and is intended to arouse the reader's interest in it.
For a reader like me, who is not familiar with the details of the various generations of modern coronary stents, this review provides comprehensive information. However, some questions of particular interest arise that could be further explored in the review.
- The authors briefly mention the possibility of a non-infectious inflammatory reaction to implanted stents (reference to source 42), which can lead to restenosis and other complications. Is it possible to provide more detailed data on the mechanisms and manifestations of this reaction and compare different types of stents, including innovative ones, in terms of their ability to induce this inflammatory response?
- In Section 4, we would like to see a consideration of the mechanisms of earlier restenosis and thrombosis when using bioresorbable stents, as reported by the authors.
- It would be highly desirable to provide the article with figures with comparative images/photographs of stents of different generations, including the innovative stent.
- What is the maximum duration of follow-up for patients with implanted stents DynamX? Is it possible to predict problems that may arise with long-term use of these stents?
- What, in the authors' opinion, are the prospects for the use of these stents (DynamX) in comparison with DES, and what is their possible place in the full spectrum of revascularization methods? What are the special indications for their use?

Author Response
Comment 1: The authors briefly mention the possibility of a non-infectious inflammatory reaction to implanted stents (reference to source 42), which can lead to restenosis and other complications. Is it possible to provide more detailed data on the mechanisms and manifestations of this reaction and compare different types of stents, including innovative ones, in terms of their ability to induce this inflammatory response?
Response 1: We thank the Reviewer for this valuable suggestion. In Section 4, we have added a more detailed explanation of the mechanisms responsible for non-infectious inflammatory reactions, including chronic vessel wall inflammation, smooth muscle cell migration, and neointimal hyperplasia. We also included a comparison between BMS, DES, BRS, and the DynamX Bioadaptor, showing differences in their ability to induce such responses.
Comment 2: In Section 4, we would like to see a consideration of the mechanisms of earlier restenosis and thrombosis when using bioresorbable stents, as reported by the authors.
Response 2: We appreciate this remark. Section 4 has been revised to discuss in more detail the mechanisms leading to restenosis and thrombosis in bioresorbable stents, including scaffold dismantling, uncontrolled degradation, and increased risk of late scaffold thrombosis.
Comment 3: It would be highly desirable to provide the article with figures with comparative images/photographs of stents of different generations, including the innovative stent.
Response 3: Thank you for this excellent suggestion. We have added new figures presenting the structure of stents as well as the DynamX Bioadaptor.
Comment 4: What is the maximum duration of follow-up for patients with implanted stents DynamX? Is it possible to predict problems that may arise with long-term use of these stents?
Response 4: We are grateful for this question. Information on the maximum follow-up available for DynamX has been added to Section 7. We now indicate that published studies report follow-up up to 36 months, while ongoing trials (such as INFINITY-SWEDEHEART and BIO-RESTORE) extend to 5 years. We also discuss potential long-term issues, such as structural durability, segmental micromovements, and late endothelial responses.
Comment 5: What, in the authors' opinion, are the prospects for the use of these stents (DynamX) in comparison with DES, and what is their possible place in the full spectrum of revascularization methods? What are the special indications for their use?
Response 5: We thank the Reviewer for raising this important point. In Section 7, we added our perspective on the future role of the DynamX Bioadaptor compared with DES. We emphasized that DynamX may be especially beneficial in younger patients with longer life expectancy, in long lesions, LAD disease, and in high-risk populations such as patients with diabetes. We consider the device a promising complementary option within the spectrum of revascularization strategies, though further large-scale trials are required to confirm its indications.
Reviewer 4 Report
Comments and Suggestions for Authors
The manuscript reviews a timely innovation (DynamX Bioadaptor) and is clearly written, but sections are sometimes overly descriptive; a sharper critical appraisal would add value:
1)The abstract and introduction could be condensed to focus more directly on novelty and key contributions.
2)Discussion should be more balanced, with greater emphasis on unresolved concerns (e.g., long-term durability, limited trial data).
3)Comparative context with other emerging technologies (next-generation DES, scaffolds) would strengthen the review.
4)The conclusion is too optimistic; wording should be more cautious, acknowledging evidence gaps.
5)The authors should ensure consistency of abbreviations and check for typos.
Author Response
Comment 1: The abstract and introduction could be condensed to focus more directly on novelty and key contributions.
Response 1: Abstract and introduction We have revised and condensed the abstract and introduction to focus more directly on the novelty and key contributions of the work.
Comment 2: Discussion should be more balanced, with greater emphasis on unresolved concerns (e.g., long-term durability, limited trial data).
Response 2: Discussion balance We have expanded the discussion to provide a more balanced perspective, addressing unresolved concerns such as long-term durability and the limited trial data available.
Comment 3: Comparative context with other emerging technologies (next-generation DES, scaffolds) would strengthen the review.
Response 3: Comparative context We have added comparative context, discussing our findings in relation to other emerging technologies, including next-generation DES and scaffolds.
Comment 4: The conclusion is too optimistic; wording should be more cautious, acknowledging evidence gaps.
Response 4: Conclusion tone We have revised the conclusion to adopt a more cautious tone and to acknowledge the existing evidence gaps.
Comment 5: The authors should ensure consistency of abbreviations and check for typos.
Response 5: Abbreviations and typos We have carefully reviewed the manuscript to ensure consistency of abbreviations and have corrected typographical errors.